# Emergence of a short peptide based reductase via activation of the model hydride rich cofactor

Ayan Chatterjee[1], Surashree Goswami[1], Raushan Kumar ⓘ[1], Janmejay Laha[1] & Dibyendu Das ⓘ[1]✉

In extant biology, large and complex enzymes employ low molecular weight cofactors such as dihydronicotinamides as efficient hydride transfer agents and electron carriers for the regulation of critical metabolic processes. In absence of complex contemporary enzymes, these molecular cofactors are generally inefficient to facilitate any reactions on their own. Herein, we report short peptide-based amyloid nanotubes featuring exposed arrays of cationic and hydrophobic residues that can bind small molecular weak hydride transfer agents (NaBH$_4$) to facilitate efficient reduction of ester substrates in water. In addition, the paracrystalline amyloid phases loaded with borohydrides demonstrate recyclability, substrate selectivity and controlled reduction and surpass the capabilities of standard reducing agent such as LiAlH$_4$. The amyloid microphases and their collaboration with small molecular cofactors foreshadow the important roles that short peptide-based assemblies might have played in the emergence of protometabolism and biopolymer evolution in prebiotic earth.

Early versions of metabolism used simpler ways of harvesting energy from the environment by exploiting energetically downhill transfer of electrons from low molecular weight cofactors. This in turn possibly fueled the generation of proton gradient required for all important anhydride pyrophosphate synthesis[1,2]. Such early means of chemical energy production had to be assisted by primitive catalysts, which facilitated the minimal metabolic cycles (catabolism and anabolism) through a series of connected oxidoreduction reactions. These biochemical transformations gradually contributed towards the complexities required for the threshold of the emergence of protocellular life[3]. In contemporary biology, low molecular weight negatively charged cofactors such as NADPH, and NADH play vital roles as hydride transfer agents and energy carriers for critically important metabolic processes such as glycolytic activation, oxidative phosphorylation, insulin secretion, and so forth[4,5]. However, these small molecular cofactors necessitate the presence of three-dimensional binding pockets offered by the highly evolved enzymes and their

electronic environment to carry out hydride transfers that are otherwise not feasible with only cofactors outside of proteins. The complex binding pockets featuring arginine-rich cationic residues for Coulombic interactions, bind and activate the negatively charged cofactors to carry out metabolically relevant electron transfer (oxidation/reduction) reactions[6]. However, there remains a critical gap in the understanding of how primitive crowding of numerous elementary cofactors and minimal biopolymers facilitated the early oxidoreduction processes leading to the emergence of protometabolism, compartmentalization with concomitant evolution of structurally complex biocatalysts[7–20]. Towards this end, short peptides proficient of accessing inherent repetitive cross β amyloid phases with solvent-exposed amino acid residues, have been often argued to be convenient surrogates of primitive protein folds[21–30]. Oligomeric peptide fragments with a limited number of amino acid residues can access robust nanoconstructs featuring antiparallel stacks of β-sheets under harsh environmental conditions[31–38]. We asked whether such short peptide

[1]Department of Chemical Sciences and Centre for Advanced Functional Materials, Indian Institute of Science Education and Research (IISER) Kolkata, Mohanpur 741246, India. ✉e-mail: dasd@iiserkol.ac.in

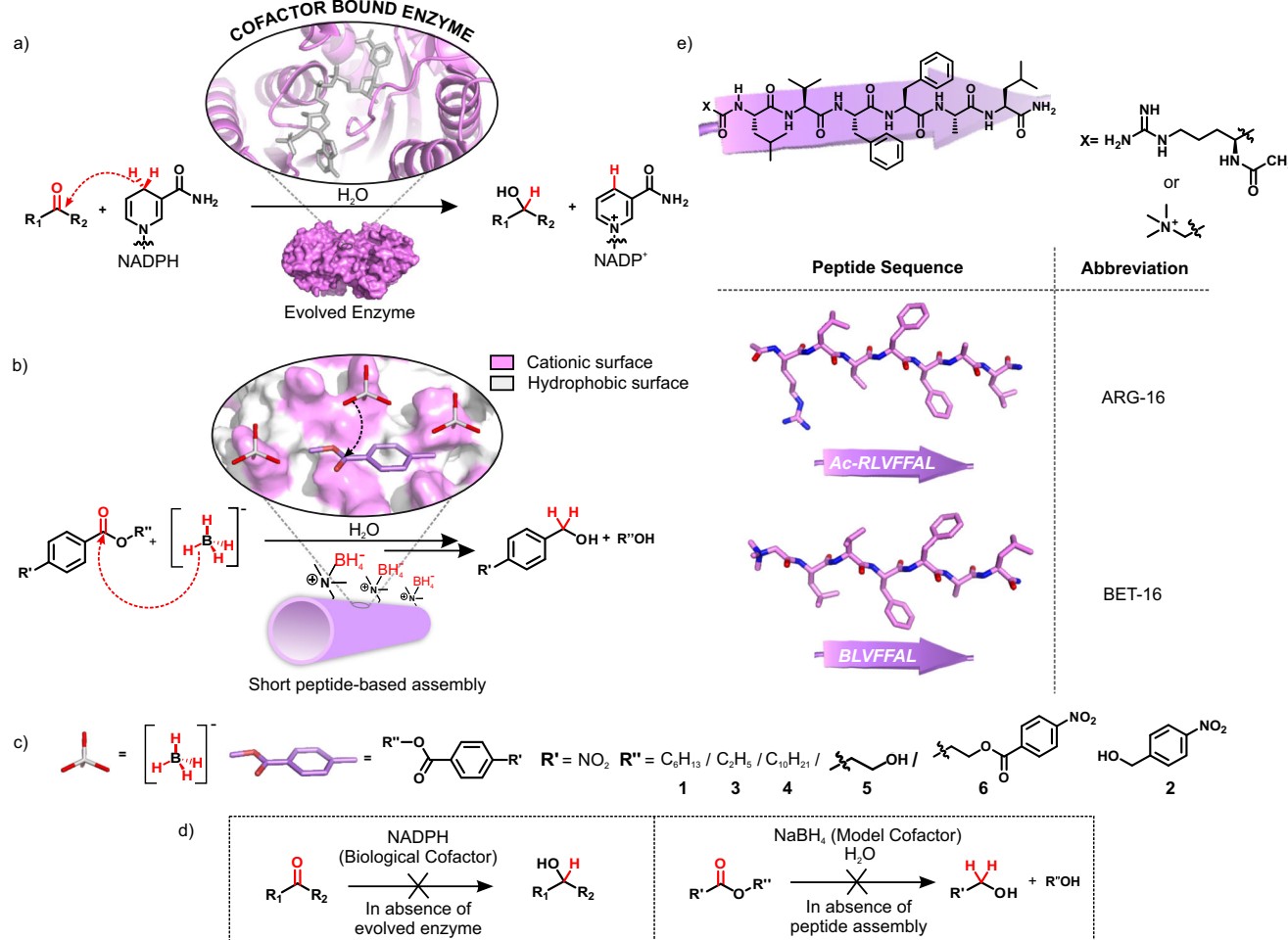

**Fig. 1 | Comparison of catalytic pathways between biological and model systems.** Schematic representation of reduction reaction using **a** modern enzyme-bound cofactor NADPH, **b** short peptide-based nanostructures with surface-bound model cofactor NaBH₄. **c** Chemical structures of the model cofactor, the substrates used, and the products. **d** Representation of reduction reaction using only cofactors in aqueous milieu. **e** Chemical structures, extended β-strand configurations, and abbreviations of the cationic peptide sequences.

fragments could exploit electrostatic effects via its binding sites punctuated with charged residues to carry out hydride transfer with rudimentary cofactors to facilitate chemical transformations that are otherwise disfavoured in the aqueous milieu.

Herein, we show short amyloid-based nanotubes that can capitalize its surface exposed arrays of proximally aligned cationic residues with hydrophobic patches for colocalization of substrates along with a weak hydride transfer agent (NaBH₄) to carry out reductions of esters to alcohols in water (Fig. 1). Just as seen for NADPH/NADH, in the absence of the peptide assemblies, the model hydride donor (NaBH₄) is unable to reduce the substrate molecules (Fig. 1a, b, d). Such synergistic action of multiple residues, small molecular guests, and the utilization of an electronic environment despite the short sequence underpins the potential of short peptide-based amyloid assemblies for their possible role in the emergence of protometabolism and biopolymer evolution.

## Results

### Selection of peptide sequence

The short peptide fragment ([17]LVFF[21]A) from the nucleating core of Aβ (1-42) amyloid was used. This sequence is extensively studied as it is seen in the fibrillar deposits of Alzheimer's disease[39,40]. The pentapeptide core fragment from 17−21 residues of Aβ upon mutation is known to access distinct amyloid morphologies with acute responsiveness to environment switches[39–41]. Specific condensations on this nucleating core were done with arginine and leucine at the N- and C-termini, respectively, resulting in the generation of Ac-[16]RLVFFA[22]L-NH₂ (ARG-16, Fig. 1e). Peptide ARG-16 self-assembled to form homogenous nanotubes with dimension 40 ± 4.2 nm as observed under transmission and scanning electron microscopy (TEM, SEM, Fig. 2a, c, Supplementary Fig. 1, 2 suggested hollow nanotubular morphologies, CD and FTIR showed the characteristic β-sheet signature, Supplementary Fig. 3). The peptide strands arranged in antiparallel out-of-register β-sheet stacks with the N-terminal cationic residues and the C-terminal hydrophobic leucine residues positioned outside the H-bonded β-sheet arrays to access well-defined surfaces of nanotubes[39,40]. The addition of Thioflavin-T (ThT), which is a marker for amyloid aggregates, provided a strong fluorescence signal at 416 nm upon addition to peptide nanotubes (Fig. 2b). To probe the binding efficiency of these microphases for small molecules, the assemblies were incubated with a fluorescent dye (FITC) as a model hydrophobic guest. Confocal micrographs demonstrated fluorescent assemblies that suggested surface-bound fluorophore molecules and highlighted the localizing capabilities of ARG-16 assemblies towards small molecular guests (Fig. 2f). Addition of negatively charged gold nanoparticles (GNP_neg, zeta potential of −24.9 ± 12.8 mV, Supplementary Fig. 4) to ARG-16 resulted in uniform arrays of nanoparticles on the nanotubular surfaces, underpinning the presence of solvent-exposed cationic arginine residues to maintain the colloidal stability (Supplementary Fig. 5). The cationic surface of the nanotubes was further

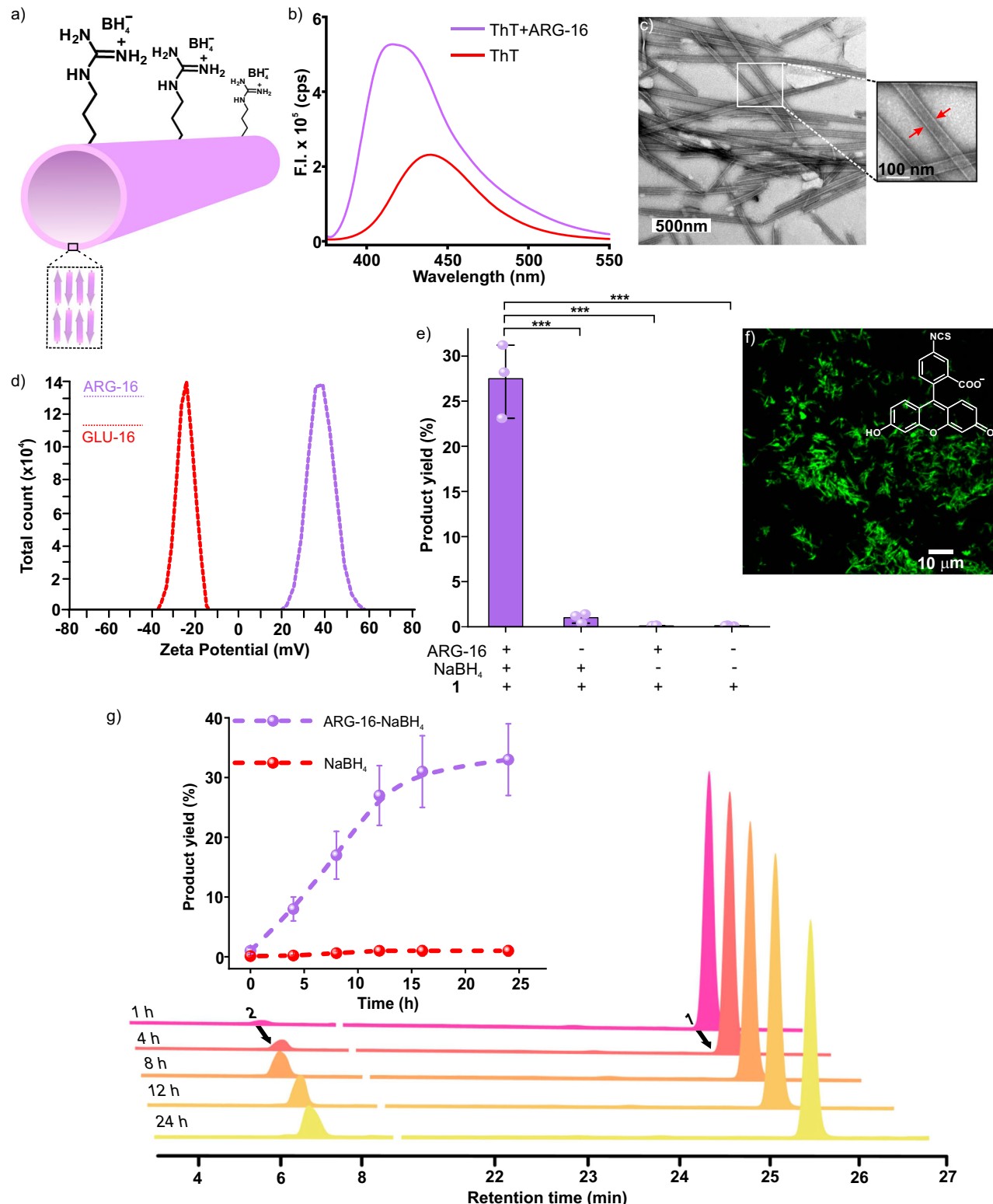

**Fig. 2 | Characterization and catalytic action of ARG-16. a** Schematic representation of ARG-16 nanostructures, **b** fluorescence spectra of ThT in the presence and absence of ARG-16, **c** TEM micrograph of ARG-16, inset: zoomed portion shows the hollow interior of ARG-16 nanotubes indicated by red arrows. The experiment was repeated for at least three times. **d** Representative zeta potential plot of ARG-16 and GLU-16 assemblies (the experiment was repeated for three times), **e** bar diagram of the yield of alcohol (**2**) by system and controls in 12 h, **f** CLSM micrograph of FITC bound ARG-16, **g** Time-dependent HPLC traces of reaction mixture of reduction, inset showing time course reaction yield by ARG-16-NaBH₄ system. The error bars are calculated from three independent experiments. All data are presented as the mean ± s.d. (*n* = 3 separate experiments). For the bar diagram, statistics were estimated using a two-sided homoscedastic 't' test (*$p \leq 0.05$, **$p \leq 0.01$, ***$p \leq 0.001$; exact *p* value for NaBH₄-**1** system is 0.00037, ARG-16-**1** system is 0.00031 and for the system with only **1** is 0.00031) in comparison with the ARG-16-NaBH₄-**1** system as reference. Source data are provided as a Source Data file.

probed via zeta potential measurements which showed high positive zeta potential value for these amyloid assemblies (38.5 ± 2 mV, Fig. 2d). Notably, in presence of GNP$_{neg}$, the zeta potential of ARG-16 reduced to 20.9 ± 3.5 mV, which indicated the effective binding of the negatively charged gold nanoparticles by the cationic assemblies (Supplementary Fig. 6).

## Catalytic Reduction

In extant biology, non-protein cofactors (NADPH) that are otherwise weak hydride donors, are activated via binding with oxidoreductases such as aldo-keto reductases for catalytic reductions of proximally bound substrates. Towards this end, the templating capabilities of the short peptide-based assemblies of ARG-16 to localize a model 'weak' hydride transfer agent (NaBH$_4$) was explored. As the terminal polar (arginine) and nonpolar (leucine) residues came together to form a surface that feature both hydrophobicity and arrays of cationic charges, the peptide nanotubes were expected to enhance the local interfacial concentrations of non-protein model cofactor, NaBH$_4$, and activate it for reductions in aqueous milieu. In addition, the hydrophobic binding patches on the amyloid surfaces would also help to bind and colocalize hydrophobic substrates (esters) and thus lower the activation barrier required for the hydride transfer[32,40]. Hence, a model hydrophobic substrate, hexyl 4-nitrobenzoate (1) was used (Fig. 1c). Indeed, the ARG-16 nanotubes (0.5 mM) were found to promote reduction of 1 (1 mM) in water to generate the product 4-nitrobenzyl alcohol (2) in presence of NaBH$_4$ (10 mM, Fig. 2e, g, Supplementary Fig. 7, 41, product 2 was collected from HPLC and characterized from mass and NMR spectra, Supplementary Fig. 7, see detailed reaction procedure in Supplementary Information, pH of the reaction mixture was 10)[42]. The time course reduction profile demonstrated a saturation region in product formation after 12 h (Fig. 2g inset, product yield of 27.5 ± 4% in 12 h, control with NaBH$_4$ did not result in any measurable product formation). It is important to note that NaBH$_4$ is a very weak hydride transfer agent and thus unable to reduce ester molecules in the presence of aqueous and organic media[43]. At slightly lower pH 9.5, ARG-16 showed even higher product conversion of ca. 39.1 ± 2.5%, indicating the presence of higher cationic surfaces at lower pH (Supplementary Fig. 8). Further lowering the pH to 9, led to the degradation of the cofactor NaBH$_4$ due to the evolution of H$_2$ gas, resulting in no noticeable product yield (at physiological pH no reduction was observed, Supplementary Fig. 9). At higher pH-12, no product generation could be observed possibly due to the neutralization of the assemblies and their colloidal instability. To check the role of morphologies, ARG-16 nanotubes were disassembled using hexafluoroisopropanol (HFIP, Supplementary Fig. 10a)[41]. This disassembled system in the presence of NaBH$_4$ showed a meagre yield of 2.2 ± 0.1% for the reduction of 1 (Supplementary Fig. 10c). Further, Na$_2$SO$_4$ was added to ARG-16 to induce bundling of the peptide nanotubes where the reaction yield was drastically reduced to 3.6 ± 0.1% due to the loss of colloidal stability of the binding surfaces (Supplementary Fig. 10b, c)[44]. Interestingly, peptide sequence Ac-$^{16}$RLVFFA$^{22}$R-NH$_2$ (ARG-16ARG-22) with an additional arginine residue at 22$^{nd}$ position ($^{22}$L to R mutation) did not show characteristic cross-β signature peaks in CD spectroscopy (Supplementary Fig. 11). Moreover, when ThT dye was incubated with the ARG-16ARG-22 assemblies, the characteristic increase in the emission intensity of the dye was not observed (Supplementary Fig. 12). The peptide sequence assembled into nanofibrillar morphologies as observed under AFM and TEM (Supplementary Fig. 13a, 14a). However, ARG-16ARG-22 could not effectively promote the reduction of 1 in the presence of NaBH$_4$ and showed drastically lower yield (ca. 8.0 ± 0.3% in 12 h, Supplementary Fig. 15) compared to ARG-16. This indicated that the highly cationic sequence ARG-16ARG-22, lacking cross-β arrangement, failed to effectively localize the hydrophobic lipidated esters. Further, the pentameric hydrophobic core 'LVFFA' was scrambled and modified to 'FLVFA' leading to the

isomeric sequence Ac-$^{16}$RFLVFA$^{22}$L-NH$_2$ (ARG-16PHE-17). These peptide assemblies did not show characteristic cross-β packing signatures and formed nanofibrillar morphologies (suggested by CD spectra, ThT binding assay, AFM and TEM images, Supplementary Fig. 11, 12, 13b, 14b). In the presence of ARG-16PHE-17, significantly lower yield (ca. 14.5 ± 0.5% in 12 h) compared to ARG-16 was observed (Supplementary Fig. 15). A shorter tripeptide, Ac-$^{18}$RF$^{20}$F-NH$_2$ (ARG-18PHE-20) was also synthesised. The tripeptide did not access nanotubular morphologies upon assembly and also did not demonstrate the characteristic cross-β sheet signature (from CD, ThT spectra, AFM, and TEM images, Supplementary Fig. 11, 12, 13c, 14c). No noticeable product conversion was observed when this tripeptide was used in the reaction system (ca. 0.12 ± 0.02% in 12 h, Supplementary Fig. 15). Importantly, from Powder X-Ray diffraction (PXRD) studies, ARG-16 showed an intense peak at d-spacing of ca. 10.54 Å (Supplementary Fig. 16) corresponding to the distance of β-sheet laminates while the controls ARG-16PHE-17 and ARG-18PHE-20 with modified hydrophobic 'LVFFA' core did not feature any prominent peaks in that region. This further suggested the strong propensity of the paracrystalline antiparallel cross-β stacks of 'LVFFA' core of ARG-16 to effectively bind hydrophobic esters, which is lacking in the mutants ARG-16PHE-17 and ARG-18PHE-20[32,40]. Notably, the absence of nanotubular morphologies or well-resolved nanostructures (ARG-16ARG-22, ARG-16PHE-17 and ARG-18PHE-20, Supplementary Fig. 13, 14) could be an additional factor for the poor activity of the mutated peptides. In another control sequence, arginine was mutated with glutamic acid congener to yield Ac-$^{16}$ELVFFA$^{22}$L (GLU-16), which showed a mixture of tubular and sheet-like morphologies (Supplementary Fig. 17). The exposed negative charges on the amyloid surfaces were validated from negative zeta potential value (−27.5 ± 1 mV, Fig. 2d) and the loss of GNP$_{neg}$ binding capability (Supplementary Fig. 18). However, the yield of the ester reduction plummeted to a meagre 3 ± 1% (Fig. 3a, Supplementary Fig. 19). These results underlined that the short peptide-based assemblies can selectively bind and activate weak hydride transfer agents for the emergence of catalytic function and emulates the traits of extant enzymes.

## Supercharged peptide framework and catalytic augmentation

Inspired by Nature's ingenuity which uses post-translational methylation of amino acids in advanced proteins as a strategy to generate cationic residues for catalytic or regulatory functions[45], we asked if the short peptide-based assemblies could expose such kinetically locked cationic surfaces (quaternized amines) as opposed to guanidinium groups (ARG-16) which are pH dependent. This could in principle result in an increased affinity towards negatively charged cofactor molecules. In this context, the arginine residue was mutated with a quaternized glycine (betaine trimethyl ammonium salt, $^{16}$BLVFFA$^{22}$L, BET-16) at the N-termini (Fig. 1e). The sequence $^{16}$BLVFFA$^{22}$L (BET-16) upon assembly formed hollow nanotubular morphologies as observed under TEM (Fig. 3b) and atomic force microscopy (AFM, Fig. 3c). The latter indicated BET-16 nanotubes with an average height of 7.8 ± 1.6 nm (Fig. 3a, c). CD spectra showed a negative peak at 224 nm suggesting the presence of β-sheet secondary conformation (Fig. 3d, inset shows FTIR spectra indicating antiparallel cross-β-sheet signature). The binding capabilities of BET-16 were further probed using ThT dye which showed an increase in the fluorescence emission intensity upon binding to the assemblies (Supplementary Fig. 20). Further, binding of GNP$_{neg}$ to BET-16 nanotubes indicated the presence of cationic nanosurfaces (Fig. 3e, g, zeta potential of BET-16 was measured to be 43.4 ± 2 mV, Supplementary Fig. 21). CLSM micrographs of fluorophore bound peptide networks indicated the strong binding propensity of the nanostructures towards different hydrophobic guest molecules such as Coumarin 343, FITC and RITC (Fig. 3f, Supplementary Fig. 22).

Next, the reducing efficacies of BET-16 assemblies with NaBH$_4$ towards the substrate 1 was investigated. The reaction yield marked a

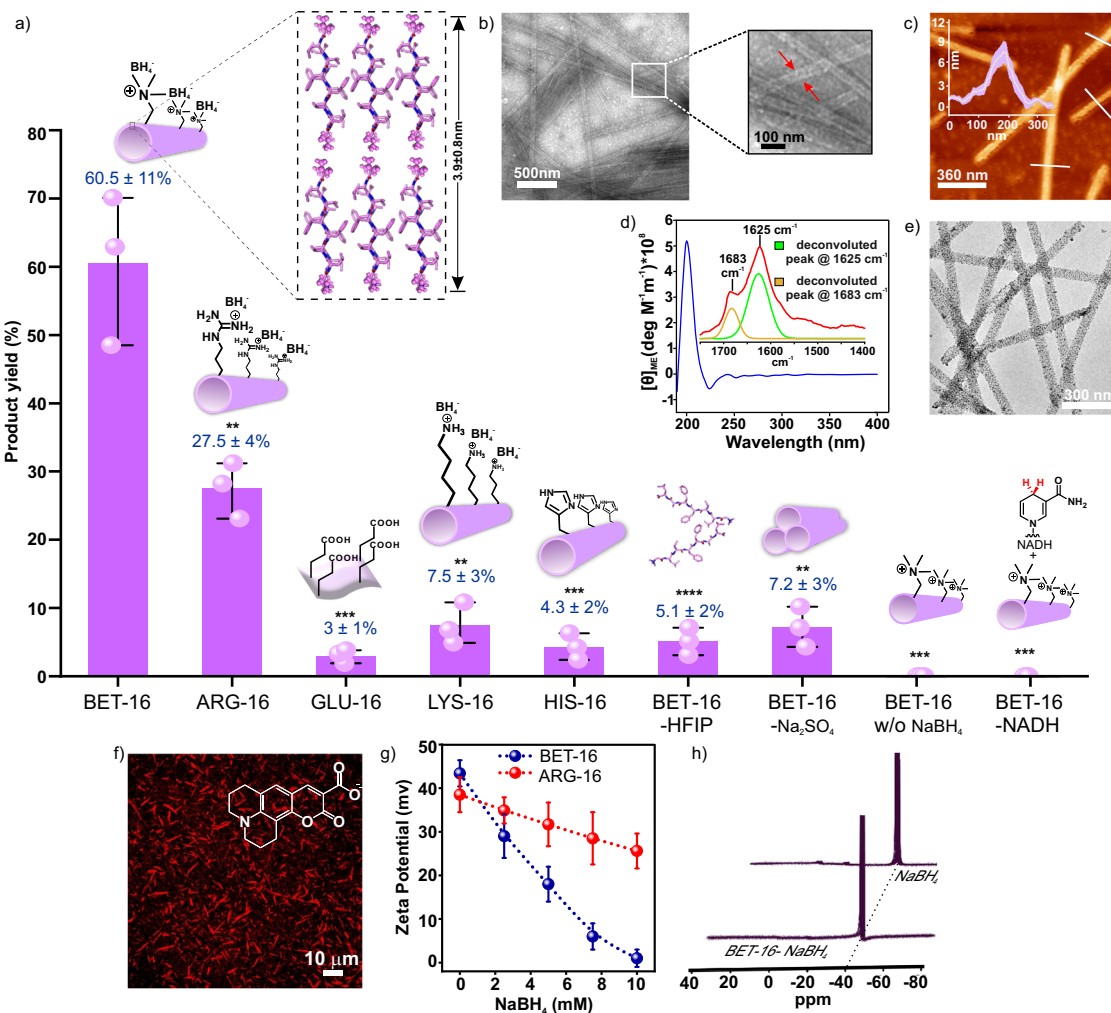

**Fig. 3 | Detailed rate comparison between different systems. a** Bar diagram of the product (**2**) yield in the presence of different systems at 12 h, **b** TEM micrograph of BET-16, inset: zoomed portion shows the hollow interior of BET-16 nanotubes indicated by red arrows, **c** AFM image of BET-16 with line profile (used for showing the height of the bilayer in the dotted box). **d** CD and FTIR (deconvoluted at amide I region, between 1600 and 1700 cm⁻¹) profile (inset) of BET-16 assemblies, **e** TEM micrograph of GNP$_{neg}$ bound BET-16 nanotubes, **f** CLSM micrograph of Coumarin 343 bound BET-16. All the microscopic experiments were repeated for at least three times. **g** Zeta potential of BET-16 in presence of variable NaBH₄ concentration,

**h** normalized $^{11}$B NMR profile of NaBH₄ in presence and absence of BET-16. The error bars are calculated from three separate experiments. Data are presented as the mean ± s.d. ($n = 3$ independent experiments). For the bar diagram, statistics for all control systems were estimated using two-sided homoscedastic 't' tests in comparison with the BET-16 system taken as reference (*$p \leq 0.05$, **$p \leq 0.01$, ***$p \leq 0.001$; exact $p$ value for ARG-16 system is 0.0082, GLU-16 is 0.0008, LYS-16 is 0.0012, HIS-16 is 0.0009, BET-16-HFIP is 0.0010, BET-16-Na₂SO₄ is 0.0012, BET-16 without NaBH₄ is 0.0006 and BET-16-NADH system is 0.0006). Source data are provided as a Source Data file.

striking improvement from ca. 27.5 ± 4% (for ARG-16) to ca. 60.5 ± 11% in 12 h at pH 10, reinforcing the importance of supercharged amyloid surfaces exposed with quaternary ammonium residues for the efficient reduction of simple esters in aqueous milieu (Fig. 3a, Supplementary Fig. 23–25, 41). The time-dependent product generation for BET-16 assemblies showed a similar trend as observed for ARG-16 assemblies (see Supplementary Fig. 24, 25 for detailed kinetic analysis). To get further insights, the Coulombic interaction between the negatively charged cofactors (BH₄⁻ ion) with cationic BET-16 assemblies was examined. We anticipated that the electrostatic interactions between exposed quaternary ammonium residues with the anionic BH₄⁻ ions would be much higher for BET-16 and hence with the same cofactor concentration, the reduction of surface potential would be expectedly more pronounced for BET-16 in comparison to arginine (conjugated acid form) exposed ARG-16 nanotubes. Indeed, upon gradual addition of NaBH₄ to BET-16, the zeta potential values of the nanostructures were significantly reduced. In contrast, the surface potential for ARG-16 was lowered to 25 ± 6 mV at 10 mM of cofactor concentration, indicating greater cofactor binding potential of BET-16 (Fig. 3g).

Notably, upon increasing the concentration of NaBH₄ to 30 mM, the zeta potential (1.4 ± 0.43 mV, Supplementary Fig. 26) and product yield of ARG-16 reached a similar value (ca. 57.7 ± 0.2%, Supplementary Fig. 27) as observed for BET-16. At lower cofactor concentration (5 mM), the product yield was further reduced to 12.5 ± 3% (Supplementary Fig. 27). These results indeed suggested that the cofactor localization ensured a hydride-rich amyloid surface for the efficient ester reduction. The reduction by BET-16 assemblies displayed maximum yield at pH 10, suggesting the optimal condition for activating the weak hydride transfer agent (NaBH₄, Supplementary Fig. 28, no reduction was observed at physiological pH, Supplementary Fig. 29)[42]. $^{11}$B NMR spectroscopy was performed to examine the stability of the cofactor NaBH₄ in presence of BET-16 (Fig. 3h). NaBH₄ was incubated with the BET-16 assemblies for 1.5 h and was subsequently centrifuged to obtain the cofactor bound peptide pellet. The supernatant was discarded to remove unbound NaBH₄. Subsequently, $^{11}$B NMR spectra of the borohydride-bound redispersed peptide pellet was recorded. The $^{11}$B peak at −41.5 ppm of the borohydride in the redispersed pellet matched with the $^{11}$B peak of free NaBH₄. This supported the presence

of cofactor $NaBH_4$ and also its stability on the BET-16 assemblies[46]. In addition, energy dispersive X-ray spectra (SEM-EDS) of $NaBH_4$ bound BET-16 assemblies in a scanning electron microscope was performed which showed the presence of element boron (B) on the BET-16 assemblies suggesting the effective co-localization of the cofactor on the peptide surfaces (see Method section for the experimental details, Supplementary Fig. 30). Further, to gain more insight into the effect of bound cofactor on the product conversions, $NaBH_4$ was incubated with the BET-16 assemblies for 30 min followed by centrifugation. The pellet was redispersed in water to obtain the preformed complex of cofactor-bound peptide assemblies. The reaction of this preformed complex with **1** resulted in a slight decline in the product yield ($52.1 \pm 6.5\%$) in 12 h, presumably due to the fact that the unbound excess $NaBH_4$ in solution drove the equilibrium towards a hydride-rich amyloid surface (Supplementary Fig. 31). In presence of the biological cofactor NADH, BET-16 assemblies, however, failed to reduce **1**. This result indicated that in comparison to $NaBH_4$, the significantly larger NADH cofactor with sugar and nucleobase moieties was presumably unable to bind to the surface in a productive conformation required for reduction (Fig. 3a, Supplementary Fig. 32). For controls, the 16th residue (arginine) was mutated with lysine and histidine generating the sequences Ac-16KLVFFA22L (LYS-16) and Ac-16HLVFFA22L (HIS-16) which formed identical nanotubular morphologies with similar diameters (Supplementary Fig. 33). These assemblies were expected to be close to neutral at the alkaline reaction medium (lysine with $pK_a$ of ca. 10.5 and histidine with ca. 6.5). The reaction yields were found to be modest $7.5 \pm 3\%$ and $4.3 \pm 2\%$ for LYS-16 and HIS-16, respectively at pH 10, which were likely due to the lack of highly cationic surfaces as present in BET-16 microphases (Fig. 3a, Supplementary Fig. 34). At pH 9.5, LYS-16 showed almost ca. 2-fold higher product conversion (ca. $14.9 \pm 5.2\%$, Supplementary Fig. 8) than at pH 10 which could be due to the presence of higher cationic charges on the surfaces of LYS-16 assemblies at pH 9.5. HIS-16 showed almost similar yield at pH 9.5 compared to pH 10, possibly due to the neutral condition of the assemblies in this pH regime ($3.9 \pm 2.2\%$, Supplementary Fig. 8). For further insight into the role of hydrophobic cross-β grooves of amyloid nanostructures, BET-16 was treated with HFIP to disassemble the peptide nanotubes (Supplementary Fig. 35a). Notably, the reaction yield for disassembled BET-16 plummeted to only $5.1 \pm 2\%$ (Fig. 3a, Supplementary Fig. 35b). Further, due to treatment with $Na_2SO_4$ (known for bundling capabilities towards nanotubular structures)[44], bundled BET-16 resulted in marginal product formation ($7.2 \pm 3\%$) that could be attributed to the reduced availability of the binding surfaces (Fig. 3a, Supplementary Fig. 36). In absence of the cofactor $NaBH_4$, no product generation could be observed by BET-16 (Fig. 3a, Supplementary Fig. 37). In addition, BET-16 showed an order of magnitude higher (ca. 12-fold) product yield compared to the previously reported cationic micellar aggregates of cetyl-trimethylammonium bromide (CTAB) (ca. $5.2 \pm 3\%$ yield, Supplementary Fig. 38)[14,15]. Furthermore, by changing the counterion of borohydride to use tetrabutylammonium borohydride (TBAB), the product yield (ca. $7.6 \pm 3.2\%$) was ca. 8-fold lower than the BET-16 system, thus underpinning the high efficacies of short peptide-based assemblies in the activation of weak hydride transfer reagents towards water-mediated ester reduction (Supplementary Fig. 38).

## Reusability and substrate selectivity

Reusability is an important aspect of a catalytic/templating system for various biotechnological and industrial applications. At this point, we were interested to see whether these short peptide-based self-assemblies could be recycled for the catalytic reduction. Briefly, the BET-16 peptides were centrifuged, followed by washing after each reaction cycle of 12 h, to separate the assemblies from the reaction mixture. Importantly, the reduction competences of BET-16 nanotubes were maintained up to three consecutive cycles with product yield of $42.3 \pm 7\%$ (Fig. 4a). The robust performance and recyclability of the short peptide-based systems despite the vigorous stirring, high pH, and multiple batches of hydrophobic substrate additions are advantageous attributes, not only from the context of potential industrial applications but also argue their candidacies as early catalysts under harsh prebiotic conditions (TEM micrograph in Supplementary Fig. 39 showed the retention of structural integrity under such harsh reaction conditions). Substrate variation with this minimal amyloid-based system was done with two more substrates containing different side chain lengths (ethyl-(4)-nitrobenzoate, **3** and decyl-(4)-nitrobenzoate, **4**, Fig. 1c). Among these, BET-16 was found to promote the reduction of **1** ($60.5 \pm 11\%$) most efficiently compared to **3** ($41.5 \pm 5\%$) and **4** ($50.5 \pm 7\%$, Fig. 4b, Supplementary Fig. 42). The difference of affinity for reduction of **3** and **4** substrates was modest yet a trend in terms of the hydrophobicity and optimal chain length versus yields could be observed. In contrast, when a comparatively polar substrate (hydroxyethyl (4)-nitrobenzoate, **5**) was used, a significantly reduced yield of $9 \pm 3\%$ was obtained (Fig. 4b, Supplementary Fig. 40, 43a). This was expected as surface of the nanotubes would not be able to effectively localize the highly polar ester (**5**).

## Controlled reduction

Encouraged by the critical roles of hydrophobic binding pockets for inducing selective substrate localization, BET-16 in principle, was expected to demonstrate acutely controlled reduction. For this ethane-1,2-diol was condensed with 4-nitrobenzoic acid to obtain 2-((4-nitrobenzoyl)oxy)ethyl 3-nitrobenzoate (**6**) and this ester was exposed to the BET-16-cofactor system. Indeed, the reduction of the bis-ester (**6**) by the BET-16 assemblies was found to be highly controlled with the generation of corresponding monoester **5** ($46.4 \pm 7\%$) and the alcohol **2** ($49.1 \pm 5\%$) after 12 h of reaction time (Fig. 4c, Supplementary Fig. 43b). In contrast, a standard reducing agent such as $LiAlH_4$ failed to demonstrate any significant degree of controlled reduction as it reduced both the ester moieties of **6** (Fig. 4c) in 12 h. The reason for the precise single-site reduction by the BET-16-cofactor system stemmed from the generation of the polar intermediate (**5**), which impeded its binding on the amyloid surface and restricted any further reduction. This underpinned the abilities of the short peptide-based supercharged amyloid nanotubes to carry out efficient, selective as well as controlled reduction reaction in the aqueous milieu (Fig. 4c).

## Discussion

Contemporary biology exploits a host of small molecular cofactors, which are usually incapable to efficiently catalyse any reactions on their own. These cofactors need to be activated via binding with highly complex three-dimensional binding pockets of extant proteins to facilitate the diverse sets of biochemical transformations. Herein, we demonstrate the templating potential of minimal peptide-based self-assembled scaffolds capable of binding and activating small molecular weak hydride transfer agents (cofactors), to facilitate kinetically challenging ester reduction reactions in aqueous milieu. Short peptide-based amyloid phases featuring exposed arrays of cationic and hydrophobic residues were able to bind and activate locally concentrated anionic cofactors $NaBH_4$ for efficient reduction of lipidated ester substrates. In addition, the paracrystalline amyloid phases showed substrate selectivity as it could perform controlled ester reduction in water. This work builds on the capabilities of short amyloid-based catalytic assemblies and their collaboration with small molecular cofactors foreshadowing the diverse catalytic roles of cofactor-bound complex biocatalysts. This contributes to our growing understanding of the emergence of complex catalytic biopolymers in early earth and looking beyond, it can contribute towards the development of advanced functional nanomaterials that could be adaptable for industrial applications.

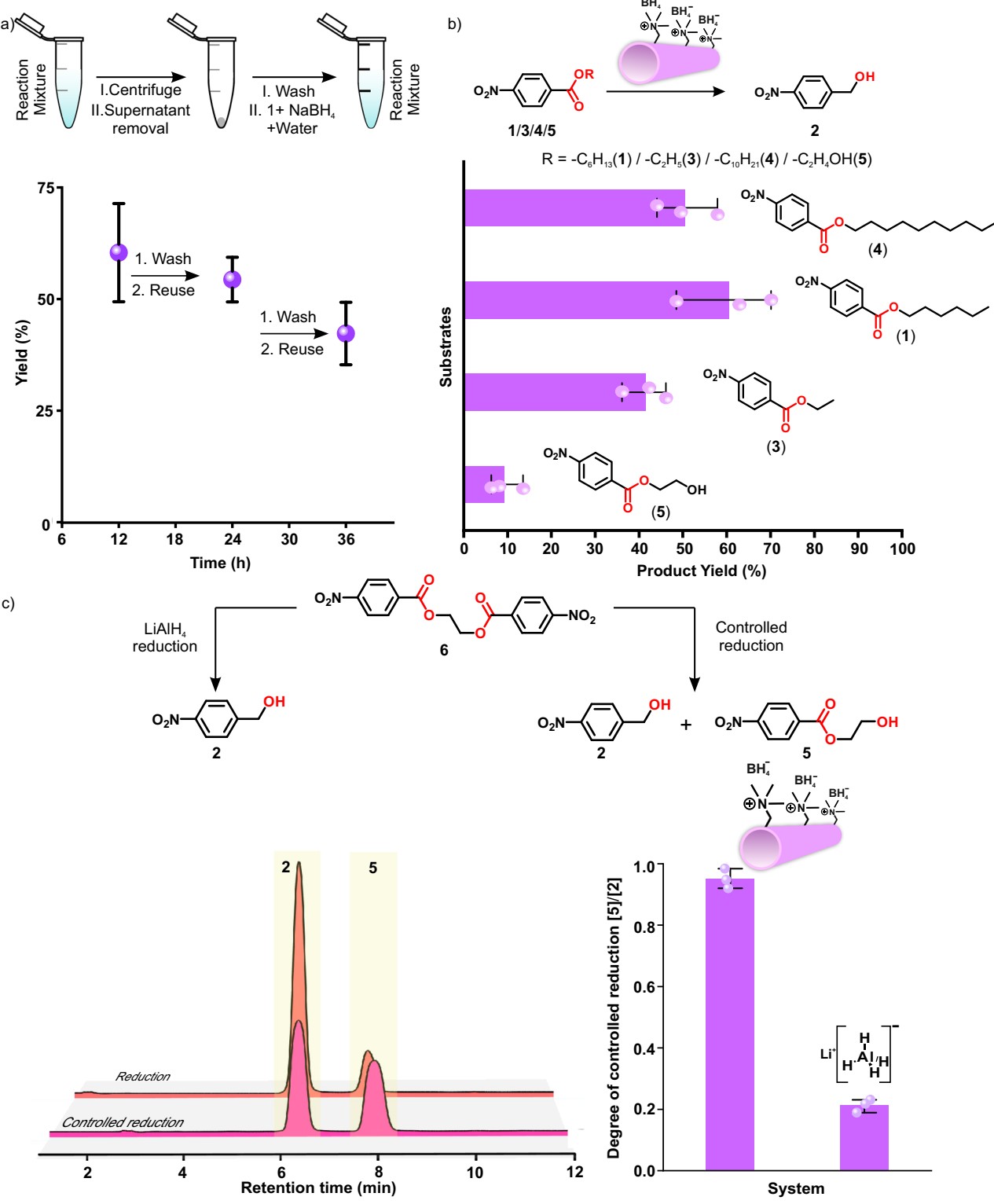

**Fig. 4 | Controlled reducing ability of short peptide and model cofactor combination. a** Variation of product yield during the recycling process of BET-16, **b** substrate selectivity by BET-16, **c** reaction scheme, HPLC chromatogram, and the bar diagram of the degree of controlled reduction of **6** by BET-16-NaBH₄ and LiAlH₄.

The error bars are calculated from three separate experiments. Data are presented as the mean ± s.d. ($n = 3$ independent experiments). Source data are provided as a Source Data file.

## Methods

The activator DIC (N,N-diisopropylcarbodiimide), all fluorenylmethyloxycarbonyl (Fmoc) protected amino acids, DIPEA (*N,N*-diisopropylethylamine), piperidine, trifluoroacetic acid (TFA), 1,1,1,3,3,3-Hexafluoro-2-propanol (HFIP), gold chloride trihydrate,

trisodium citrate, Thioflavin T, sodium borohydride and β-Nicotinamide adenine dinucleotide (NADH) were purchased from Sigma Aldrich. Oxyma was purchased from Nova Biochem. 4-Nitrobenzoic acid and Glycine betaine were purchased from SRL. Imidazole acetic acid was purchased from Alfa Aesar. Hexanol, ethanol,

decanol, ethylene glycol and HEPES (4-(2-hydroxyethyl)-1-piper-azineethanesulfonic acid) were purchased from TCI, Japan. All solvents and Fmoc-Rink amide MBHA Resin were purchased from Merck. Milli-Q water was used throughout the experiments.

## Peptide assembly

Lyophilized powder of purified peptides (amount required to make 1 mL stock solution of 2.5 mM concentration) was added to 1 mL of 40% (v/v) acetonitrile-water containing 0.1% trifluoroacetic acid and continuously mixed for ~2 min under vortex[40]. The homogeneous solution was kept at 4 °C for two months. The formation of the assembly was monitored through CD for β-sheet signature, and the homogenous population of nanostructures was checked under TEM.

## Redispersion technique

After the assembly stage was completed, 1 mL of the assembled peptide solution (2.5 mM concentration, 40% (v/v) acetonitrile-water containing 0.1% trifluoroacetic acid as solvent) was taken in 1.5 mL microcentrifuge tube and centrifuged for 15 min at 12,000 rpm (rotor F45-30-11) at 4 °C temperature in Eppendorf centrifuge 5804 R[40]. After discarding the supernatant, the pellet was redispersed in an equal volume of water (1 mL) to maintain the 2.5 mM of concentration for overall system. This stable homogenous stock solution of redispersed peptide assemblies (2.5 mM) was used for all the experiments in this work.

## Transmission electron microscopy (TEM)

To check the self-assembly before redispersion, 10 μL of peptide samples (150 μM, diluted from 2.5 mM stock solution) aged in 40% (v/v) ACN/water containing 0.1% TFA, was dropcasted on a TEM grid for 3 min. The excess solution was wicked up by using filter paper, followed by incubation with 3 μL of 1% (w/v) uranyl acetate for 3 min. The excess staining agent was then removed by filter paper. The samples were further dried in vacuo for a few hours before imaging (TEM image of ARG-16 is provided in Supplementary Fig. 1).

For the water-redispersed peptide assemblies, 10 μL of peptide sample (150 μM) was dropcasted on a TEM grid for 3 min. The excess solution was wicked up by using filter paper, followed by incubation with 3 μL of 1% (w/v) uranyl acetate for 3 min. The excess staining agent was removed by filter paper and the samples were further dried in vacuo for few hours before imaging.

For HFIP or $Na_2SO_4$ treated assemblies, 150 μM of the disassembled or bundled peptide assemblies, respectively was used. 10 μL of the peptide sample was dropcasted on the TEM grid for 3 min and then the excess solution was wicked up using filter paper and then 3 μL of 1% (w/v) uranyl acetate was added and kept for 3 min which was then removed using a filter paper. The samples were then dried in vacuo for few hours before imaging.

To probe the morphology of the BET-16 assemblies in the presence of $NaBH_4$ after two cycles of reduction reactions with **1**, 30 μL of the reaction mixture at pH 10 ([BET-16] = 500 μM, [$NaBH_4$] = 10 mM, [**1**] =1 mM) was taken and diluted with 20 μL water. 10 μL of this solution was then dropcasted on a TEM grid for 3 min followed by the addition of uranyl acetate as mentioned before. After that samples were placed under vacuum in a desiccator. TEM micrographs were recorded with a JEOL JEM 2100 with a Tungsten filament at an accelerating voltage of 200 KV.

## Scanning electron microscopy/energy dispersive spectroscopy (SEM-EDS)

Scanning electron microscopy (SEM) in combination with energy dispersive X-ray spectroscopy (SEM-EDS) were recorded using a Carl Zeiss SUPRA 55VP instrument and an Oxford Instruments X-Max system with INCA software, respectively. For imaging, the samples were prepared following the similar procedure used for TEM imaging. For the SEM-EDS analysis, the BET-16 assemblies (0.5 mM) were incubated with $NaBH_4$ (1 mM) for 30 min, followed by centrifugation. The pellet formed was redispersed in water, and 20 μL of the solution was dropcasted on a silicon wafer and dried in vacuo for few hours before imaging. Elemental mapping of boron and sodium on the BET-16 nanotubes was performed by excluding the highly intense peaks of the elements carbon, oxygen, nitrogen, and silicon for the precise detection of the specific peaks of boron and sodium on the sample. In another experiment, the pellet of $NaBH_4$-bound BET-16 assemblies obtained after centrifugation, was lyophilized and this lyophilized powder was further used for the elemental mapping.

## Gold nanomaterials binding studies

200 μL of gold colloid (negatively charged $GNP_{neg}$, 250 μM) were separately introduced to 5 μL of the aqueous dispersions of peptide assemblies (150 μM) for studying the gold nanoparticle binding. The obtained mixture was kept for incubation for ca. 2 h at room temperature until a purple-red precipitate was formed followed by centrifugation. The pellet was redispersed in water. 10 μL of this solution was carefully drop casted on TEM grids and kept for 2 min. The extra solution was wicked off by using the filter paper.

## Thioflavin T assay

The redispersed peptide assemblies of ARG-16, ARG-16ARG-22, ARG-16PHE-17, ARG-18PHE-20 and BET-16 assemblies (each concentration of 100 μM) were mixed with ThT (30 μM) in water by gentle tapping using pipette, followed by an incubation for 30 min. Thereafter, solution was taken in a fluorescence quartz cuvette of 10 mm pathlength. The fluorescence spectra were recorded with Horiba Fluoromax spectrofluorometer at 25 °C using excitation at 370 nm.

## Circular dichroism

JASCO J-810 circular dichroism spectrometer with a Peltier temperature controller was used to record CD spectra at a constant temperature of 25 °C. The water-redispersed peptide assemblies (concentration of 0.5 mM) were placed into a quartz cuvette of 1 mm pathlength. Each spectrum was obtained by scanning wavelengths from 400 nm to 190 nm at a scanning rate of 100 nm/min. Three successive wavelength scans were taken to average for each sample.

## Fourier-transform infrared spectroscopy

IR spectra were collected using a Bruker IR spectrometer after sample aliquots were dried into a thin film (model no: Alpha) in ATR mode (Platinum ATR) at room temperature with 256 scans at 4 $cm^{-1}$ resolution. Background spectra were subtracted from each sample spectrum.

## Powder X-ray diffraction (PXRD)

For the PXRD experiments, samples were prepared by centrifugation of the peptide assemblies (0.5 mM). After discarding the supernatant, the pellet formed was lyophilized and the PXRD measurements were performed on the lyophilized peptide powder with a Rigaku (mini flex II, Japan) powder X-ray diffractometer having Cu Kα = 1.54059 Å radiation.

## HFIP disassembly and $Na_2SO_4$ bundling studies

For disassembling the peptides with HFIP, 1 mL of redispersed peptide assemblies (2.5 mM) were centrifuged for 15 min at 10000 rpm at 4 °C in a 1.5 mL microcentrifuge tube to form pellet. After discarding the supernatant, equal volume of HFIP was added to disassemble the nanostructures by incubation for 2 h. Afterwards, HFIP was removed by $N_2$ blowing and dried for 2 h under vacuum. An equal volume of water was added to the remnant disassembled peptide to make the final concentration to 2.5 mM. Next, 1 mM of **1** was added to 500 μM of disassembled peptide (diluted from disassembled peptide stock

solution of concentration 2.5 mM) followed by NaBH$_4$ (10 mM) addition. The reaction mixture was then monitored in HPLC after 12 h. For bundling of peptide nanotubes, aqueous solution of Na$_2$SO$_4$ (500 μM) was added to redispersed peptide assemblies (500 μM) for 2 h followed by the addition of **1** (1 mM) and NaBH$_4$ (10 mM). The final reaction mixture at 12 h was monitored via HPLC analysis.

## Confocal microscopy

1 μL of Coumarin 343, FITC, and RITC (150 μM) dyes were mixed separately with the redispersed peptide solution (2.5 mM) and incubated for 2 h. The solution was then casted on a glass slide and enclosed with a cover glass. The images were recorded in Olympus Laser Scanning Confocal System Model FV3000 (part of the Atomic Force Microscope with Rheological Measurement and Confocal Imaging Unit Facility, supported by Swarnajayanti (SB/SJF/2020-21/08)) at a frame of 512 × 512, with 12-bit depth with a 100X, with the laser line of 488 nm and 561 nm.

## Zeta potential measurement

Zeta potential was measured using a Malvern Zetasizer Nano ZS (Malvern, UK). Briefly, redispersed peptide assemblies (0.5 mM) were incubated with different concentration of NaBH$_4$ for 1 h and the samples were probed through Zetasizer. Notably, the zeta potential of peptide assemblies (ARG-16/BET-16) without NaBH$_4$ were found to be almost similar in both aqueous or pH 10 buffered medium. For the zeta potential measurements of negatively charged GNP$_{neg}$, the concentration was taken to be 125 μM. To check the zeta potential of GNP$_{neg}$ bound ARG-16 assemblies, 500 μL, 125 μM of GNP$_{neg}$ (used from a stock solution of GNP$_{neg}$ of 250 μM concentration) was added to ARG-16 assemblies (500 μM) and was incubated for 2 h at room temperature (final volume of the mixture was 1 mL) until a purple red precipitate was formed. The mixture was centrifuged, and the pellet was redispersed in an equal volume of water (1 mL). Zeta potential was checked with this redispersed homogenous solution of GNP$_{neg}$ bound ARG-16 assemblies.

## NMR

To characterize the synthesized compounds, all $^1$H and $^{13}$C NMR spectra were recorded at Bruker (500 MHz) and JEOL (400 MHz) NMR spectrometers, respectively (Supplementary Figs. 44–53). For $^{11}$B NMR study, 1 mL of peptide assemblies (500 μM) were incubated with NaBH$_4$ (10 mM) for 1 h (pH 10) and then centrifuged for 5 min at 12,000 rpm. The supernatant was discarded, and the borohydride-bound peptide pellet was redispersed in D$_2$O. Afterwards, the sample was probed via $^{11}$B NMR at 500 MHz NMR spectrometer, Bruker.

## HPLC analysis

High performance liquid chromatography (HPLC) was performed in a Waters HPLC system fitted with a photodiode array detector and mass detector (Waters QDa) using X-bridge® C18 5 μm, 4.6 × 150 mm analytical column at a flow rate of 1 mL/min with a gradient from 30 % to 70% acetonitrile in water (both solvents contain 0.1% TFA) and a total run time of 30 min. The formation of reduced product **2** and reactants were extracted at 276 nm. The product yields were calculated from the equation $\left( \left[ \frac{[Product]}{[Product] + [Unreacted\ Reactant]} \right] \times 100 \right)$. The concentrations of the unreacted substrate and generated products were calculated by noting the peak areas of reaction mixture and using the standard plots (Supplementary Fig. 41–43) of substrates and products with known concentration.

## Reporting summary

Further information on research design is available in the Nature Portfolio Reporting Summary linked to this article.

## Data availability

The authors declare that all data supporting this work are contained in graphics displayed in the main text or in the Supplementary Information. Data used to generate these graphics are available from the authors upon request. Source data are provided with this paper.

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

## Acknowledgements

D.D. is thankful to SJF Grant SB/SJF/2020-21/08, CSIR (0406)/21/EMR-IIC and SERB CRG/2022/003607, GOI for financial assistance. A.C. and S.G. acknowledge CSIR, India and J.L. acknowledges UGC, India.

## Author contributions

D.D. conceived and supervised the overall project. A.C., S.G., R.K. and J.L. conceived and performed all the experiments. All authors discussed the results and commented on the manuscript.

## Competing interests

The authors declare no competing interests.
