## [Peer Review File · Nature Communications]

Emergence of a short peptide based reductase via activation of the model hydride rich cofactorREVIEWER COMMENTS

Reviewer #1 (Remarks to the Author):

This work describes the entrapment of sodium borohydride onto the short peptide-based assemblies to catalyze the reduction of ester substrates. The design of peptide is based on well-studied literatures about related peptide with similar sequence that becomes catalytic active upon self-assembly into nanotubes. The methodology of fabricating catalytic assemblies for reducing esters by entrapping borohydrides onto the nanostructures via electrostatic interaction has been reported, where sodium borohydride was immobilized at an aqueous cationic micellar surface (*Organic Letters*, 2004, 6, 4133-4136). The content of this work provides some advantages, including higher activity and certain substrate selectivity, but does not promise significant novelty on designing new artificial enzymes compared to previous reported literatures. Meanwhile, the manuscript is not well-organized, and my enthusiasm is hampered by the concerns described below.

Major concerns:

- (1) It is understandable that authors selected aromatic esters as substrates for facilitating the monitoring of catalytic rate by HPLC. However, it is well known that the aromatic esters can be slowly hydrolyzed into acids and alcohols under aqueous conditions even without catalysts, and especially, the reaction in this work was carried out at pH 10.0, which is more beneficial for the hydrolysis of esters. Although there a new peak occurs at an earlier retention time after the reaction, it is not sure whether this product is from ester reduction or hydrolysis. Additional experimental evidence, such as MS and NMR, should be provided to confirm the molecular structure of product.
- (2) The authors claim the colocalization of sodium borohydride with the peptide assemblies. This should be experimentally proved by SEM/TEM-EDS (Energy Dispersive Spectrometer).
- (3) There are some missing controls but rather important, for example, the reduction rate of esters in the studied medium in the presence of all peptide assemblies but without sodium borohydride, and the zeta potential of GNPneg and GNPneg bonded ARG-16.
- (4) The catalytic reactions are carried out at harsh condition (pH 10.0), and it seems that adding Na₂SO₄ as a control can affect the morphology of BET-16 according to the TEM image in Supplementary Fig 24, so how is the stability of peptide assemblies under reaction condition during the whole reaction time?

(5) The AFM images in supplementary Fig.5 show that peptide ARG-16PHE-17, ARG-18PHE-20 and ARG-16ARG-22 have lost the nanotube-like morphologies. CD and Th T assay should be applied in the characterization of peptide ARG-16PHE-17 and ARG-18PHE-20 beyond the PXRD to confirm their conformations, respectively.

Minor concerns:

(1) The manuscript is not well-organized, for example, the statement “This was further supported from the high positive zeta potential value for these amyloid assemblies (38.5 ± 2 mV, Fig. 2d).” was completely insular and without context; the names, ARG-22 and BET-22 in Fig. 1, never appear in the whole main text and I guess they are corresponding to ARG-16 and BET-16, respectively.

(2) The quality of some characterizations should be improved, such as CD and FTIR spectrum, and PXRD profile, and also wider width of spectral window should be provided in FTIR and PXRD to show more detailed information.

(3) The initial preparation of such system needs significant higher concentration of sodium borohydride (10 mM) than peptide ARG-16 (0.5 mM), so how is the function of the exceeded sodium borohydride and the catalytic activity of peptide with different concentration of sodium borohydride.

Reviewer #2 (Remarks to the Author):

In the current manuscript authors described their findings on catalysis of esters reduction made by a combination of short amyloid fibers together with a suitable cofactor such as sodium borohydride. In particular, they utilized a hydrophobic scaffold of amyloid fibers together with the positively charged amino acid residues not only to accommodate the substrates of the reaction but also to activate borohydride ion to efficiently reduce the substrates, which is very suppressed if the sodium borohydride is used alone in such reaction and also if peptide design is unsuitable (e.g. the peptide forms no fibers). The authors showed that the same catalytic fibers can be used for this reaction multiple times by simple centrifugation-resuspension and the addition of borohydride. The catalytic effect is dependent to a degree on the substrate polarity. This work expands the field of catalytic amyloid fibers as it adds one more reaction to the plethora of catalysis via amyloids and small cofactors.

Despite the efforts demonstrated by the authors, there are numerous aspects in the current manuscript that do not allow me to recommend publishing this manuscript in Nature Communications.

The first and foremost aspect is that the majority of peptides presented here, except for BET-16, were previously reported by the same authors and others. They were used in different reactions and it is quite surprising that authors did not cite the most recent Chemical Science paper with those compounds (<https://doi.org/10.1039/D2SC03205H>). Thus, the lack of novelty is the major problem for publishing these results in Nature Communications.

Next, I would like to point out other aspects, since the manuscript by itself requires a substantial improvement.

- Sodium borohydride is a good representative example of the cofactor, however, it would be more representative to add others, biologically relevant ones (like NADPH, which was mentioned), especially since authors connect their catalytic system with the field of Origin of Life;

- The methodology of the reduction reaction raises questions for further determination of the kinetic parameters and yields of the reaction. Here and later I refer to both the main manuscript and the Supplementary Information file. First of all, why does the "Peptide Assembly" section of the Main Text state the necessity of Acetonitrile/Water/TFA for storage? For what purposes such a homogeneous solution (probably containing no assemblies, since it has TFA) was prepared? The next section "Redispersion technique" is not connected with the previous section as it states that there is assembled peptide at the very beginning and it forms a pellet upon centrifugation. Where did the assembled peptide (with the known concentration of 2.5 mM) come from?

- From the Supplementary information file it is unclear, how were peptide solutions/assemblies prepared for application in the reaction. Was it the solid that just was mixed with water or was there any stock solution prepared in a suitable solvent? How the concentration was determined in the case of a solid being dispersed in water? How accurate can it be to state concentration 500 μ M if the assemblies were formed instantaneously by

solid dispersion in water (and not the true solution where one can indeed be sure that concentration by weight corresponds to the real concentration) according to the protocol of authors? An example of the method where these concerns were to an extent solved can be found in the paper cited by authors <https://www.nature.com/articles/nchem.1894>, where the stock solutions were prepared in highly acidic aqueous medium and proved to be real solutions, which were then used to prepare fibers by raising the pH.

- It is not explained how the substrates were introduced in the reaction mixture. Were there any stock solutions (and then what was the solvent influence if any) or the substrates were just added as solids/liquids in an attempt to redisperse them? Is this then heterogeneous catalysis (with all concerns about it), since substrates from structure seem to be highly hydrophobic unless it is proved by authors that they are soluble at the concentration used in the current study (1mM)?

- Another question is the pH control and pH adjustments in the reactions reported. There is no detailed information about that provided by the authors (except for the short caption note in SI Fig. 19). Were there any buffers? The authors report the different pH-dependent results (Supplementary Figure 19) of their reaction but there is no detailed procedure for these experiments. The same is true for the sections with CTAB and TBAB experiments. For the reduction with LiAlH_4 the lack of details makes it difficult to follow and would make it difficult to reproduce. It is the case where “Briefly” should be extended to “In details”. Borohydride itself in an aqueous medium should raise pH quite significantly; this control was not reported by the authors.

- Thioflavin T assay again raises the question of how redispersed peptides were prepared and CD measurements lack the details like concentration of the samples and where they come from.

- The authors stated that introducing a second Arg leads to highly cationic peptide ARG-16ARG-22 which is incapable of efficiently accommodating the substrates. To me, the AFM data provided (Supplementary Fig. 5c,6) to support this statement simply means the absence of fibrils-type assembly with its hydrophobic pockets and this is the reason for poor catalytic activity. The same is true for the other peptides for which AFM data is shown.

- Page 5 lines 120 – 135 contains a large array of peptides and their catalytic (un)efficiency in the form of yields. There are mistakes in it, e.g. ARG-18PHE-20 a short peptide is stated to be inactive, and the Supporting Information Figure 6 shows a yield ~20%. The same is true

for the rest of the data in these lines. The difference between the data in text and in SI raises questions about its quality and the quality of analysis.

- The pH profile for Lys-16 and His-16 (especially with Lys, which has $pK_a \sim 10.5$) catalytic activity must be present for the correct comparison of mutants with Agr-16 and BET-16. So far only one compound was presented with catalytic activity dependent on pH.
- Page 7 lines 180 – 183 is in partial disagreement with the Figure 3g. Zeta potential is indeed lowered for ARG16 to a lesser degree than for BET-16, but not slightly how authors stated. The quality of the plot does not allow the reader to justify a drop to 25 ± 6 mV from ~ 39 mV.
- Page 7 lines 192 – 194 report information about ^{11}B NMR. Does that mean the authors detected borohydride ions in the solution after the redispersion of the pellet?
- Page 7 lines 200 – 202. It has to be that position 16 was mutated to K or H, since authors started from ARG-16. The pH profile for Lys-16 and His-16 (especially with Lys, which has $pK_a \sim 10.5$) catalytic activity must be present for the correct comparison of mutants with Agr-16 and BET-16.
- It is unclear, why treatment with HFIP was done exclusively for the BET-16 and not for the ARG-16. This is just an extra setup but it proves the generality of the disassembly (and disappearance of catalytic binding sites) on catalysis. The same is true for treatment with Na_2SO_4 . The information about this experiment in detail is absent in both the main manuscript and SI.
- CTAB yield is unclear from the 3D-bar graphs provided, especially considering the comparison of the CTAB-borohydride and BET-16 yields. Although, it is clear that BET-16 gives a better result.
- Page 9 There is probably a typo for yield of substrate 4 reduction, as it is not the same as shown in Figure 4b. That means substrates 4 and 3 have practically the same yield despite the difference in the hydrophobic alkyl chain. Moreover, according to the data provided with error bars, substrates 1, 3, and 4 demonstrated quite close yields of products (this is especially prominent for substrates 3 and 1), suggesting a very minor difference in their affinity to the catalyst.
- In controlled reduction, both BET-16 and ARG-16 should be expected to have this selective behavior. If the authors would like to reduce the data shown in the main text, this effect could be at least shown in Supporting Information (reaction of ARG-16 with substrate 6).

- It is also not surprising that substrate 6 was reduced selectively to the product/substrate 2 and 5 with enhanced formation of 5, rather than only 2 since authors have already shown poor reactivity of 5. What is unclear is how this experiment was performed, since no information was presented in either the main text or Supplementary Information.

Minor comments:

- Figure 1e has the wrong abbreviation for the peptides reported.
- Alcohol 2 (nitrobenzyl alcohol) would be important to introduce in Figure 1 and not in Figure 4.
- Rapid reduction during the time course of 12 hours is rather a strong statement.
- Missing HRMS for synthesized substrates in SI

Report summary

The manuscript needs a major improvement to be published elsewhere. Methods described here require a much better and more detailed explanation: right now the key experiments are unclear and cannot be reproduced. The representation of the data needs correction: multiple aspects are missing, and Figures and captions also missing the details. Without that improvement, the conclusions of the manuscript are in insufficient agreement with the data presented.

Reviewer #3 (Remarks to the Author):

The article delves into the exploration of short peptide-based amyloid nanotubes' ability to bind and activate weak hydride transfer agents, specifically NaBH₄, making these agents more efficient in the reduction of ester substrates in water and speculates that similar molecular entities may have played in the emergence of early metabolic processes and biopolymer evolution on prebiotic Earth. They demonstrate that some, but not all these structures, with exposed arrays of specific residues, can interact with weak hydride transfer agents and substrates, enhancing their efficiency. These paracrystalline structures demonstrate recyclability, substrate selectivity, and controlled reduction and they seemed

to outperform standard reducing agents such as LiAlH_4 . The findings hint at potential applications in developing advanced functional nanomaterials suitable for industrial use, although these applications are not properly detailed. As a conclusion the article presents a perspective on the utilization of peptide-based amyloid nanotubes in facilitating reactions that molecular cofactors on their own cannot efficiently achieve. Although interesting and novel there are some methodological and theoretical issues that the authors should address before it can be published:

1) The biophysical characterization of the assemblies deviates from standard practices. The CD spectra should be shown in the far-UV region to appreciate the typical spectral minima at 217 nm. The signal should be expressed in molar ellipticity and not as mdeg. FTIR should be presented in amide I region of the spectra and the absorbance deconvoluted to appreciate the contribution of the different secondary structure components and in particular of the beta-sheet at 320-330 cm^{-1} .

2) While the amyloid microphases' abilities are highlighted, mechanistic insights into how these assemblies interact with NaBH_4 remain scarce. This is vital for a deeper understanding of the process. How are the different peptide chains sustained in the nanotube? How the large uncompensated positive charges in ARG-16 assembly are maintained without exerting a repulsion that disentangles the structure? A molecular model of the nanotubes and molecular docking simulations of the cofactor and substrate in their surfaces would allow to clarify these important questions.

3) The argument that these assemblies might mimic early assemblies in life is attractive, but not really supported by the experiments since 1) The assembly occurs in 40% of organic solvent at a very acidic pH and 2) the activity is measured at pH 10. It is not clear if these assemblies would be formed, and more important, would be active at neutral/physiological pH, this last point should be assessed experimentally.

4) The morphological characterization across different assemblies lacks uniformity. In some they utilize TEM, others AFM, and in some they employ both, leading to discrepancies in analysis.

5) There's a variation in the shapes of different assemblies; not all form standard nanotubes. Simply attributing the activity or inactivity of control peptides to their sequence or physicochemical properties, without considering their spatial orientation, seems oversimplified.

6) Related to question 3, when Arg is substituted by His or Lys the resulting structures are clearly nanotubes but not very active. The argument is that they are not significantly protonated at pH 10, if this is the underlying reason the activity of Lys-16 should be strictly dependent on the pH and a titration curve would demonstrate unequivocally this property.

Responses:

Reviewer #1 (Remarks to the Author):

This work describes the entrapment of sodium borohydride onto the short peptide-based assemblies to catalyze the reduction of ester substrates. The design of peptide is based on well-studied literatures about related peptide with similar sequence that becomes catalytic active upon self-assembly into nanotubes. The methodology of fabricating catalytic assembles for reducing esters by entrapping borohydrides onto the nanostructures via electrostatic interaction has been reported, where sodium borohydride was immobilized at an aqueous cationic micellar surface (*Organic Letters*, 2004, 6, 4133-4136). The content of this work provides some advantages, including higher activity and certain substrate selectivity, but does not promise significant novelty on designing new artificial enzymes compared to previous reported literatures. Meanwhile, the manuscript is not well-organized, and my enthusiasm is hampered by the concerns described below.

>> The points raised by the Reviewer have been addressed in following sections with additional experiments and detailed discussion.

(1) It is understandable that authors selected aromatic esters as substrates for facilitating the monitoring of catalytic rate by HPLC. However, it is well known that the aromatic esters can be slowly hydrolyzed into acids and alcohols under aqueous conditions even without catalysts, and especially, the reaction in this work was carried out at pH 10.0, which is more beneficial for the hydrolysis of esters. Although there a new peak occurs at an earlier retention time after the reaction, it is not sure whether this product is from ester reduction or hydrolysis. Additional experimental evidence, such as MS and NMR, should be provided to confirm the molecular structure of product.

>> As suggested by the Reviewer, peak eluted at 7 min was characterized via mass (line 137, page 6, Revised SI) and NMR spectra (Supplementary Fig. 7, page 11, Revised SI) which confirmed that the product was 4-nitrobenzyl alcohol (**2**). When 4-nitrobenzoic acid was injected as standard using the same method, its elution was observed at 1.9 min.

(2) The authors claim the colocalization of sodium borohydride with the peptide assemblies. This should be experimentally proved by SEM/TEM-EDS (Energy Dispersive Spectrometer).

>> In addition to the Boron NMR experiment (lines 223-230, page 8, Fig. 3h, Revised MS), we have now performed the SEM-EDS analysis to probe the co-localization of NaBH₄ on the **BET-16** assemblies. Briefly, **BET-16** assemblies were incubated with NaBH₄ for 30 min, and subsequently centrifuged to remove unbound NaBH₄. The pellet of borohydride bound **BET-16** was lyophilized and the lyophilized powder was used for the elemental mapping. SEM-EDS analysis (Supplementary Fig. 30, page 23, Revised SI) indicated the presence of elements such as boron (B), carbon (C), nitrogen (N), oxygen (O) and very less occurrence of sodium (Na) in the **BET-16**-NaBH₄ sample (discussion and detailed experimental method are included in lines 230-233, page 8, lines 366-377, page 12, Revised Manuscript) which further suggested the co-localization of the borohydride ions on the peptide nanosurfaces.

(3) There are some missing controls but rather important, for example, the reduction rate of esters in the studied medium in the presence of all peptide assemblies but without sodium borohydride, and the zeta potential of GNP_{neg} and GNP_{neg} bonded ARG-16.

>> We have now included the controls done in presence of peptides **BET-16** and **ARG-16** (which was already included in Fig. 2e) without the addition of NaBH₄. No noticeable activity was observed (Fig. 3a, 8th data point on the x-axis, page 6, Revised Manuscript). Further, zeta potential distribution profile of GNP_{neg} and GNP_{neg} bound **ARG-16** are provided in Supplementary Fig. 4,6, pages 9-10, Revised SI

(discussion in lines 94, 96-99, 108-109, pages 3-4 and detailed experimental procedure in lines 432-438, page 13, Revised Manuscript).

(4) The catalytic reactions are carried out at harsh condition (pH 10.0), and it seems that adding Na₂SO₄ as a control can affect the morphology of BET-16 according to the TEM image in Supplementary Fig 24, so how is the stability of peptide assemblies under reaction condition during the whole reaction time?

>> Divalent anions like Na₂SO₄ are known to bundle nanotubular structures while monovalent anions such as Cl⁻, NO₃⁻ are not effective (Ref: Lu, K., *et al*, *Chem. Commun.* **26**, 2729-2731 (2007)). We observed bundling as shown in Supplementary Fig. 36a, page 26, Revised SI. Hence, the bundled nanotubes offered lesser binding surface which led to the significantly diminished reductase activity (Fig. 3a, discussion in lines 256-259, page 8, Revised MS). Regarding the stability of the assemblies under reaction conditions, TEM image of **BET-16** assemblies in presence of NaBH₄ done after prolonged incubation of 24 h (completion of two reaction cycles) indicated no bundling of the peptide assemblies (Supplementary Fig. 39, page 28, Revised SI).

(5) The AFM images in supplementary Fig.5 show that peptide ARG-16PHE-17, ARG-18PHE-20 and ARG-16ARG-22 have lost the nanotube-like morphologies. CD and Th T assay should be applied in the characterization of peptide ARG-16PHE-17 and ARG-18PHE-20 beyond the PXRD to confirm their conformations, respectively.

>> As suggested by the Reviewer, CD spectra and ThT binding assays of **ARG-16PHE-17**, **ARG-18PHE-20** and **ARG-16ARG-22** have been included in the Supplementary Fig. 11,12, pages 13-14, Revised SI. From CD spectroscopy, no characteristic β -sheet signature peaks were observed for these peptide assemblies. Further, ThT dye did not show the characteristic increase in its emission intensity when incubated with these peptide assemblies. Corresponding text is included in lines 140-144, 150-153,155-157, pages 5-6, Revised Manuscript.

Minor concerns:

(1) The manuscript is not well-organized, for example, the statement “This was further supported from the high positive zeta potential value for these amyloid assemblies (38.5±2 mV, Fig. 2d).” was completely insular and without context; the names, ARG-22 and BET-22 in Fig. 1, never appear in the whole main text and I guess they are corresponding to ARG-16 and BET-16, respectively.

>> We thank the Reviewer for pointing out these oversights. The mentioned text is changed to “The cationic surface of the nanotubes was further probed via zeta potential measurements which showed high positive zeta potential value for these amyloid assemblies (38.5±2 mV, Fig. 2d)” in the text, lines 96-98, page 3, Revised Manuscript and the acronyms in Fig. 1 have been corrected.

(2) The quality of some characterizations should be improved, such as CD and FTIR spectrum, and PXRD profile, and also wider width of spectral window should be provided in FTIR and PXRD to show more detailed information.

>> The quality as well as the range of spectral windows of CD, FTIR and PXRD spectra have now been improved (Fig. 3d, page 6, Revised Manuscript, Supplementary Fig. 3,16, pages 9,16, Revised SI). Further, the FTIR spectra of the assemblies are now deconvoluted for better structural analysis.

(3) The initial preparation of such system needs significant higher concentration of sodium borohydride (10 mM) than peptide ARG-16 (0.5 mM), so how is the function of the exceeded sodium borohydride and the catalytic activity of peptide with different concentration of sodium borohydride.

>> The catalytic activity of **ARG-16** with different concentrations of NaBH₄ has been provided in Supplementary Fig. 27, page 21, Revised SI (corresponding text in lines 216-221, pages 7-8, Revised

MS). Expectedly, with increasing concentrations of cofactor led to higher activity of the **ARG-16**. Notably, the activity did not improve beyond 30 mM of NaBH₄, presumably due to the saturation of peptide surfaces by hydride rich cofactor as observed from the zeta potential value (1.44±0.43 mV, Supplementary Fig. 26, page 21, Revised SI).

Reviewer #2 (Remarks to the Author):

In the current manuscript authors described their findings on catalysis of esters reduction made by a combination of short amyloid fibers together with a suitable cofactor such as sodium borohydride. In particular, they utilized a hydrophobic scaffold of amyloid fibers together with the positively charged amino acid residues not only to accommodate the substrates of the reaction but also to activate borohydride ion to efficiently reduce the substrates, which is very suppressed if the sodium borohydride is used alone in such reaction and also if peptide design is unsuitable (e.g. the peptide forms no fibers). The authors showed that the same catalytic fibers can be used for this reaction multiple times by simple centrifugation-resuspension and the addition of borohydride. The catalytic effect is dependent to a degree on the substrate polarity. This work expands the field of catalytic amyloid fibers as it adds one more reaction to the plethora of catalysis via amyloids and small cofactors.

Despite the efforts demonstrated by the authors, there are numerous aspects in the current manuscript that do not allow me to recommend publishing this manuscript in Nature Communications.

>> We have now addressed all the aspects in a detailed manner as suggested by the Reviewer.

The first and foremost aspect is that the majority of peptides presented here, except for BET-16, were previously reported by the same authors and others. They were used in different reactions and it is quite surprising that authors did not cite the most recent Chemical Science paper with those compounds (<https://doi.org/10.1039/D2SC03205H>). Thus, the lack of novelty is the major problem for publishing these results in Nature Communications.

>> The findings in the present work are completely orthogonal from any report published thus far. Surely, the core sequence (inspired from Aβ 1-42 amyloid) has been used in numerous works by different groups of diverse fields. What sets apart the current work is the discovery of the short peptide-based scaffolds that can bind and activate borohydrides to efficiently reduce substrates which are otherwise unreacted; a novel methodology which is also acknowledged by the Reviewer in the first comment. In regard to the Chem Sci 2022 work, the paper demonstrated phospho-esterase activity (of short peptides of different sequences) which is again a completely different class of chemical transformation.

Next, I would like to point out other aspects, since the manuscript by itself requires a substantial improvement.

- Sodium borohydride is a good representative example of the cofactor, however, it would be more representative to add others, biologically relevant ones (like NADPH, which was mentioned), especially since authors connect their catalytic system with the field of Origin of Life;

>> In presence of the biological cofactor NADH, **BET-16** assemblies could not reduce the ester. In comparison to NaBH₄, the significantly larger NADH cofactor with sugar and nucleobase moieties was presumably unable to bind to the surface in a productive conformation required for reduction (lines 239-242, page 8, Revised MS, 9th data point of Fig. 3a, Supplementary Fig. 32, page 24, Revised SI).

- The methodology of the reduction reaction raises questions for further determination of the kinetic parameters and yields of the reaction. Here and later I refer to both the main manuscript and the Supplementary Information file. First of all, why does the “Peptide Assembly” section of the Main Text state the necessity of Acetonitrile/Water/TFA for storage? For what purposes such a homogeneous

solution (probably containing no assemblies, since it has TFA) was prepared? The next section “Redispersion technique” is not connected with the previous section as it states that there is assembled peptide at the very beginning and it forms a pellet upon centrifugation. Where did the assembled peptide (with the known concentration of 2.5 mM) come from?

>> Peptide assemblies were prepared by incubating in 40% (v/v) acetonitrile/water containing 0.1% trifluoroacetic acid and stored in refrigerator over long period of time for the maturation of the assemblies. This solvent system does not lead to dissolution of the peptides, instead helps the peptides to access monodisperse morphologies which is always advantageous for reproducible reactions. We have now added EM micrographs to demonstrate that indeed the peptides assemble to access homogenous assemblies in the ACN/H₂O/TFA solvent system (detailed method in lines 333-338, page 11, Revised MS, Supplementary Fig. 1, page 8, Revised SI). After aging the assembly in ACN/H₂O/TFA solvent system, these assemblies were centrifuged, and the redispersed pellet showed similar homogenous assemblies (Supplementary Fig. 1, page 8, Revised SI). Notably, this method is not new and has been reported in many works that involve ¹⁷LVFF²¹A core (for instance, Omosun, T., *et al*, *Nat. Chem.* **9**, 805-809 (2017); Kapil, N., *et al*, *Angew. Chem. Int. Ed.* **55**, 7772–7776 (2016)).

- From the Supplementary information file it is unclear, how were peptide solutions/assemblies prepared for application in the reaction. Was it the solid that just was mixed with water or was there any stock solution prepared in a suitable solvent? How the concentration was determined in the case of a solid being dispersed in water? How accurate can it be to state concentration 500 μM if the assemblies were formed instantaneously by solid dispersion in water (and not the true solution where one can indeed be sure that concentration by weight corresponds to the real concentration) according to the protocol of authors? An example of the method where these concerns were to an extent solved can be found in the paper cited by authors <https://www.nature.com/articles/nchem.1894>, where the stock solutions were prepared in highly acidic aqueous medium and proved to be real solutions, which were then used to prepare fibers by raising the pH.

>> We would like to thank the Reviewer for the lead in suggesting the protocol presented in nchem.1894. In the current work, we have used a different process of self-assembly (as reported in Omosun, T. O., *et al*, *Nat. Chem.* **9**, 805-809 (2017); Kapil, N., *et al*, *Angew. Chem. Int. Ed.* **55**, 7772–7776 (2016)). Briefly, the solid peptide samples (required to make 1 mL stock of 2.5 mM concentration) were added to 1 mL of 40% (v/v) acetonitrile-water containing 0.1% trifluoroacetic acid and continuously mixed for ~ 2 min under vortex. This led to a stable homogenous solution. This technique leads to initial dissolution of the peptides and is a great way to remove kinetically trapped states of undissolved peptides. Further, ageing the same system for two months helped in the equilibration and subsequent formation of highly homogenous and well dispersed peptide assemblies (2.5 mM concentration). After ageing, the assemblies were centrifuged and redispersed (details are mentioned in previous point). Detailed discussions of the ‘Peptide assembly’ and ‘Redispersion technique’ in method section (lines 333-346, page 11, Revised Manuscript) have been included to clarify this point.

As mentioned above, the protocols that we used are reported in many previous works. However, we would like to thank again the Reviewer for the suggestion of the protocol presented in the nchem.1894 paper. We will surely attempt this process of assembly in our subsequent works on peptide assemblies.

- It is not explained how the substrates were introduced in the reaction mixture. Were there any stock solutions (and then what was the solvent influence if any) or the substrates were just added as solids/liquids in an attempt to redisperse them? Is this then heterogeneous catalysis (with all concerns about it), since substrates from structure seem to be highly hydrophobic unless it is proved by authors that they are soluble at the concentration used in the current study (1mM)?

>> The stock solution of substrate (100 mM concentration) was made in acetonitrile and 10 μ L from this stock was added to the peptide borohydride aqueous mixture (final reaction volume of 1 mL). Thus, the final reaction mixture had 1% (v/v) acetonitrile in water. This method has now been added in lines 127-129, page 6, Revised SI.

- Another question is the pH control and pH adjustments in the reactions reported. There is no detailed information about that provided by the authors (except for the short caption note in SI Fig. 19). Were there any buffers? The authors report the different pH-dependent results (Supplementary Figure 19) of their reaction but there is no detailed procedure for these experiments. The same is true for the sections with CTAB and TBAB experiments. For the reduction with LiAlH_4 the lack of details makes it difficult to follow and would make it difficult to reproduce. It is the case where “Briefly” should be extended to “In details”. Borohydride itself in an aqueous medium should raise pH quite significantly; this control was not reported by the authors.

>> No buffer was used for these studies due to the interference of buffer counterions with the borohydride present in the reaction medium. For pH dependent reactions, respective pH of the reaction medium was obtained by adding few microlitres of NaOH/HCl solution. The pH was found to remain unaltered throughout the course of the reaction. Details of the pH dependent studies, the experimental procedure for CTAB and TBAB reagents and the reaction process in presence of LiAlH_4 have been included in the method section, lines 149-181, page 7, Revised SI.

- Thioflavin T assay again raises the question of how redispersed peptides were prepared and CD measurements lack the details like concentration of the samples and where they come from.

>> We thank the Reviewer for pointing this out. Now the details for CD measurements have been added (lines 391-396, page 12, Revised Manuscript). Details of redispersed peptides have been explained above and has been included in Revised Manuscript (lines 339-346, page 11).

- The authors stated that introducing a second Arg leads to highly cationic peptide ARG-16ARG-22 which is incapable of efficiently accommodating the substrates. To me, the AFM data provided (Supplementary Fig. 5c,6) to support this statement simply means the absence of fibrils-type assembly with its hydrophobic pockets and this is the reason for poor catalytic activity. The same is true for the other peptides for which AFM data is shown.

>> Indeed, the absence of well-defined nanostructures for highly cationic peptide **ARG-16ARG-22** sequence observed under AFM and TEM images, could be an additional factor (now mentioned at lines 165-167, page 6, Revised MS) for lower catalytic activity (Supplementary Fig. 13,14, pages 14-15, Revised SI). Further, PXRD, CD and ThT binding assay experiments (Supplementary Fig. 16,11,12, pages 13-14,16, Revised SI) highlighted the lack of well-defined cross β binding grooves to localize the substrates effectively by **ARG-16ARG-22** assemblies.

- Page 5 lines 120 – 135 contains a large array of peptides and their catalytic (un)efficiency in the form of yields. There are mistakes in it, e.g. ARG-18PHE-20 a short peptide is stated to be inactive, and the Supporting Information Figure 6 shows a yield ~20%. The same is true for the rest of the data in these lines. The difference between the data in text and in SI raises questions about its quality and the quality of analysis.

>> This was an oversight, and we thank the Reviewer for pointing out. The corresponding axis of the bar diagram has now been corrected (Supplementary Fig. 15, page 15, Revised SI). The ~20% yield was for the scrambled peptide ($\text{Ac-}^{16}\text{RFLVFA}^{22}\text{L-NH}_2$, **ARG-16-PHE-17**) and not for **ARG-18PHE-20** ($\text{Ac-}^{18}\text{RF}^{20}\text{F-NH}_2$) which was inactive.

- The pH profile for Lys-16 and His-16 (especially with Lys, which has pKa ~ 10.5) catalytic activity must be present for the correct comparison of mutants with Agr-16 and BET-16. So far only one compound was presented with catalytic activity dependent on pH.

>> The pH profiles of **LYS-16** and **HIS-16** have been included (Supplementary Fig. 8, page 11, Revised SI). Expectedly, marginal catalytic values were observed for both of these peptide assemblies at higher pH (pH~12) due to the absence of significant cationic charges of the peptide nanosurfaces. For **LYS-16**, higher yield was observed at pH 9.5 compared to 10 possibly due to the higher cationic charges on the peptide assemblies, as expected from lysine's pKa (ca. 10.5). Notably, pH value below 9.5 could not be achieved due to the hydrolysis of NaBH₄ in aqueous medium (Schlesinger, H. I., *et al*, *J. Am. Chem. Soc.* **75**, 215-219 (1953)). **HIS-16** showed almost similar yield at pH 9.5 (3.9±2.2%, Supplementary Fig. 8, page 11, Revised SI) compared to pH 10, possibly due to the neutral condition of the assemblies in this pH regime, as anticipated from the pKa of histidine (ca. 6.5). The discussion is included in lines 249-253, page 8, Revised MS.

- Page 7 lines 180 – 183 is in partial disagreement with the Figure 3g. Zeta potential is indeed lowered for ARG16 to a lesser degree than for BET-16, but not slightly how authors stated. The quality of the plot does not allow the reader to justify a drop to 25 ± 6 mV from ~39 mV.

>> The sentence has been revised in Revised Manuscript (lines 214-216, page 7).

- Page 7 lines 192 – 194 report information about ¹¹B NMR. Does that mean the authors detected borohydride ions in the solution after the redispersion of the pellet?

>> The redispersion of the peptide assemblies was done to remove unbound borohydride ions which are otherwise water soluble. Hence, the borohydrides that are bound to the peptides were observed from the ¹¹B NMR experiment (corresponding text in lines 223-230, page 8, Fig. 3h, Revised MS).

- Page 7 lines 200 – 202. It has to be that position 16 was mutated to K or H, since authors started from ARG-16. The pH profile for Lys-16 and His-16 (especially with Lys, which has pKa ~ 10.5) catalytic activity must be present for the correct comparison of mutants with Agr-16 and BET-16.

>> The sentence has been revised (now at line 243, page 8, Revised Manuscript). The pH profile of **ARG-16**, **LYS-16** and **HIS-16** is included in Supplementary Fig. 8, page 11, Revised SI.

- It is unclear, why treatment with HFIP was done exclusively for the BET-16 and not for the ARG-16. This is just an extra setup but it proves the generality of the disassembly (and disappearance of catalytic binding sites) on catalysis. The same is true for treatment with Na₂SO₄. The information about this experiment in detail is absent in both the main manuscript and SI.

>> As suggested by the Reviewer, we have now added the experiments of the reduction of **1** by disassembled and bundled **ARG-16** peptides using HFIP and Na₂SO₄ respectively in presence of NaBH₄. Both the disassembled and bundled **ARG-16** systems showed very low product yields compared to the cofactor NaBH₄ bound **ARG-16** assemblies, thereby suggesting the role of catalytic binding surfaces for the effective reduction of esters in water (see text in lines 134-139, page 5, Revised MS, method section in lines 408-419, page 13, Supplementary Fig. 10, page 12, Revised SI).

- CTAB yield is unclear from the 3D-bar graphs provided, especially considering the comparison of the CTAB-borohydride and BET-16 yields. Although, it is clear that BET-16 gives a better result.

>> This figure has now been revised (Supplementary Fig. 38, page 27, Revised SI) to clearly depict the yields.

- Page 9 There is probably a typo for yield of substrate 4 reduction, as it is not the same as shown in Figure 4b. That means substrates 4 and 3 have practically the same yield despite the difference in the hydrophobic alkyl chain. Moreover, according to the data provided with error bars, substrates 1, 3, and 4 demonstrated quite close yields of products (this is especially prominent for substrates 3 and 1), suggesting a very minor difference in their affinity to the catalyst.

>> This was an oversight and we thank the Reviewer. The bar diagram has now been corrected (Figure 4b, page 9, Revised MS). Indeed, the difference of affinity is modest yet a trend in terms of the hydrophobicity and optimal chain length of the substrate vs yields could be observed (corresponding text included in lines 287-289, page 10, Revised MS).

- In controlled reduction, both **BET-16** and **ARG-16** should be expected to have this selective behavior. If the authors would like to reduce the data shown in the main text, this effect could be at least shown in Supporting Information (reaction of **ARG-16** with substrate 6).

>> Indeed, the **ARG-16** is expected to demonstrate similar selection of reduction towards substrate **6**. However, considering the best yields that were observed with the novel peptide, **BET-16** was only used. We believe that the inclusion of **ARG-16** data would not add any fundamentally new aspect to the Manuscript and hence, this experiment was not performed.

- It is also not surprising that substrate 6 was reduced selectively to the product/substrate 2 and 5 with enhanced formation of 5, rather than only 2 since authors have already shown poor reactivity of 5. What is unclear is how this experiment was performed, since no information was presented in either the main text or Supplementary Information.

>> The experimental details of reduction procedure for substrate **6** have been included in the Revised SI (lines 165-181, page 7).

Minor comments:

-Figure 1e has the wrong abbreviation for the peptides reported.

>> The abbreviations have been corrected.

- Alcohol 2 (nitrobenzyl alcohol) would be important to introduce in Figure 1 and not in Figure 4.

>> The structure of alcohol **2** is now included in Fig. 1.

- Rapid reduction during the time course of 12 hours is rather a strong statement.

>> The term 'rapid' has been omitted (lines 121-122, page 5, Revised MS).

- Missing HRMS for synthesized substrates in SI

>> MS data for synthesized compounds have been incorporated.

Reviewer #3 (Remarks to the Author):

The article delves into the exploration of short peptide-based amyloid nanotubes' ability to bind and activate weak hydride transfer agents, specifically NaBH₄, making these agents more efficient in the reduction of ester substrates in water and speculates that similar molecular entities may have played in the emergence of early metabolic processes and biopolymer evolution on prebiotic Earth. They

demonstrate that some, but not all these structures, with exposed arrays of specific residues, can interact with weak hydride transfer agents and substrates, enhancing their efficiency. These paracrystalline structures demonstrate recyclability, substrate selectivity, and controlled reduction and they seemed to outperform standard reducing agents such as LiAlH_4 . The findings hint at potential applications in developing advanced functional nanomaterials suitable for industrial use, although these applications are not properly detailed. As a conclusion the article presents a perspective on the utilization of peptide-based amyloid nanotubes in facilitating reactions that molecular cofactors on their own cannot efficiently achieve. Although interesting and novel there are some methodological and theoretical issues that the authors should address before it can be published:

>> We appreciate the Reviewer's comment 'interesting and novel' and we have addressed the Reviewer's points below.

1) The biophysical characterization of the assemblies deviates from standard practices. The CD spectra should be shown in the far-UV region to appreciate the typical spectral minima at 217 nm. The signal should be expressed in molar ellipticity and not as mdeg. FTIR should be presented in amide I region of the spectra and the absorbance deconvoluted to appreciate the contribution of the different secondary structure components and in particular of the beta-sheet at 320-330 cm^{-1} .

>> As suggested, we have re-recorded the CD spectra of the peptide assemblies in wider wavelength range and as suggested, the unit 'mdeg' has been modified to molar ellipticity (Figure 3d, page 6, Revised MS, Supplementary Fig. 3, 11, pages 9, 13, Revised SI). Further, deconvoluted FTIR spectra of amide I region (between 1600 and 1700 cm^{-1}) are now included in the Revised MS and SI (Fig. 3d inset, page 6, Revised MS, Supplementary Fig. 3, page 9, Revised SI).

2) While the amyloid microphases' abilities are highlighted, mechanistic insights into how these assemblies interact with NaBH_4 remain scarce. This is vital for a deeper understanding of the process. How are the different peptide chains sustained in the nanotube? How the large uncompensated positive charges in ARG-16 assembly are maintained without exerting a repulsion that disentangles the structure? A molecular model of the nanotubes and molecular docking simulations of the cofactor and substrate in their surfaces would allow to clarify these important questions.

>> The structural analysis of peptide nanotubes featuring ' $^{17}\text{LVFF}^{21}\text{A}$ ' core have been explored previously by various techniques, from solid state NMR to SAED and diverse microscopic techniques (Lu, K., *et al*, *J. Am. Chem. Soc.* **125**, 6391–6393 (2003); Mehta, A. K., *et al*, *J. Am. Chem. Soc.* **130**, 9829–9835 (2008)). The peptide strands arrange in antiparallel out-of-register β -sheet stacks with the N-terminal cationic residues as well as the C-terminal hydrophobic leucine residues positioned outside the H-bonded β -sheet arrays to access well-defined surfaces of nanotubes. Further, the interactions between different cofactors/substrates with such assemblies have been probed through molecular docking studies by different groups (Murray, K. A., *et al*, *Proc. Natl. Acad. Sci.* **119**, e2206240119, (2022); Childers, W. S., *et al*, *J. Am. Chem. Soc.* **131**, 10165–10172 (2009); Omosun, T. O., *et al*, *Nat. Chem.* **9**, 805-809 (2017)). We speculate that the negatively charged cofactors interact with the cationic residues whereas the substrates are able to bind with the cross- β grooves for effective catalytic conversion. At this stage, although the present manuscript does not include molecular dynamics studies, our current efforts are aimed at the understanding of such interactions of the cofactor sodium borohydride and substrate with the peptide nanotubular surfaces. The corresponding discussion have been included in Revised Manuscript (lines 85-87, page 3 and lines 115-118, page 5).

3) The argument that these assemblies might mimic early assemblies in life is attractive, but not really supported by the experiments since 1) The assembly occurs in 40% of organic solvent at a very acidic pH and 2) the activity is measured at pH 10. It is not clear if these assemblies would be formed, and

more important, would be active at neutral/physiological pH, this last point should be assessed experimentally.

>> Indeed, the conditions used for assemblies in the Manuscript may not be prebiotically feasible, yet our study demonstrate the remarkable potential of short peptides to access highly active catalytic surfaces for facilitating reductions which are otherwise not feasible. It would be important to mention here that 40% (v/v) acetonitrile/water containing 0.1% trifluoroacetic acid was used to incubate the peptide assemblies. After aging, the assemblies were centrifuged to discard the supernatant containing organic solvent and the pellet was redispersed in water. The redispersed peptide assemblies were also stable at neutral pH. In this context, there are natural enzymes which work at high pH, for instance the oxidoreductase enzyme, catalase with working pH range of 7-11. The reason for using high pH for the current studies was the fact that NaBH₄ (10 mM) in the aqueous medium intrinsically elevated the pH of the system to 10. Further, if the pH of the system was kept lower than 9.5, NaBH₄ gets hydrolyzed (Schlesinger, H. I., *et al*, *J. Am. Chem. Soc.* **75**, 215-219 (1953)). As suggested by the Reviewer, we studied the reaction at neutral pH which expectedly did not yield any products (Supplementary Fig. 9, 29, pages 12,22, Revised SI, corresponding text in lines 131-133, page 5, line 221-223, page 8).

4) The morphological characterization across different assemblies lacks uniformity. In some they utilize TEM, others AFM, and in some they employ both, leading to discrepancies in analysis.

>> In addition to AFM, TEM images are now included for better clarity of structural analysis in Revised SI (Supplementary Fig. 14, page 15).

5) There's a variation in the shapes of different assemblies; not all form standard nanotubes. Simply attributing the activity or inactivity of control peptides to their sequence or physicochemical properties, without considering their spatial orientation, seems oversimplified.

>> We agree that indeed the absence of nanotubular morphologies or well-resolved nanostructures could be an additional factor for the poor activity of the mutated peptides **ARG-16ARG-22**, **ARG-16PHE-17** and **ARG-18PHE-20** apart from the absence of characteristic β -sheet signatures of these assemblies (Supplementary Fig. 11-16, pages 13-16, Revised SI). As mentioned above, our current efforts are aimed at the understanding of such spatial interactions via molecular dynamics studies. We have now included text in lines 144-145, 150-153, 155-157, 165-167, pages 5-6 of the Revised Manuscript.

6) Related to question 3, when Arg is substituted by His or Lys the resulting structures are clearly nanotubes but not very active. The argument is that they are not significantly protonated at pH 10, if this is the underlying reason the activity of Lys-16 should be strictly dependent on the pH and a titration curve would demonstrate unequivocally this property.

>> We have now provided the pH profile of **LYS-16** (Supplementary Fig. 8, page 11, Revised SI). At pH 9.5, it showed highest activity which gradually declined at higher pH due to loss of cationic charges as a result of significant deprotonation of the primary amines of lysines (pK_a~10.5). Notably, pH lower than 9.5 could not be achieved due to hydrolysis of sodium borohydride in aqueous medium (Schlesinger, H. I., *et al*, *J. Am. Chem. Soc.* **75**, 215-219 (1953)). The discussion is included in lines 249-251, page 8, Revised MS.

REVIEWERS' COMMENTS

Reviewer #1 (Remarks to the Author):

The authors have addressed all my concerns and made corrections of the manuscript. I have no comments any more, and recommend the revised manuscript for publication.

Reviewer #2 (Remarks to the Author):

In this revised version of the manuscript, the majority of the issues and concerns presented in the first report were successfully resolved. In its current state, the revised manuscript obtained much stronger support for the conclusions made by the authors. The revised manuscript is much more detailed, it is more understandable methodologically and also in general. The remaining concerns are rather minor and do not affect the overall statements made in this manuscript.

[Editorial Note: Please note that Reviewer 1 has assessed your responses to Reviewer 3 and considers them sufficiently addressed.]